# Mobile health intervention for promotion of eye health literacy

**Indra Prasad Sharma**[1,2]*, **Monica Chaudhry**[2], **Dhanapati Sharma**[3], **Raju Kaiti**[4]

**1** GKCW National Eye Center, JDW National Referral Hospital, Thimphu, Bhutan, **2** Department of Optometry and Vision Science, Amity University Haryana, Gurugram, India, **3** Department of English, Gedu College of Business Studies, Chukha, Bhutan, **4** Department of Optometry, Nepal Eye Hospital, Tripureshwor, Nepal

* indrapsharma@gmail.com

## Abstract

### Purpose

Improving eye health awareness in the underserved population is a universal eye health priority. The ubiquity of cell phones and internet usage provides new and innovative opportunities for health promotion. This study evaluated the effect of mobile health intervention (text message link) to promote eye health literacy (EHL) of priority ocular morbidities.

### Methods

This study was an intervention evaluation and employed a two-armed pre-test post-test approach. Baseline assessment on EHL was performed on 424 university students. Participants were categorised into intervention and control groups, using the 1:1 allocation ratio. The intervention and control group received a text message alone and text message with a link, respectively. EHL was assessed via a self-administered questionnaire. The primary outcome measures were changes in EHL scores between baseline and one month post-intervention. Descriptive analysis was performed to assess the cost-effectiveness of the intervention.

### Results

With low attrition and a response rate of 95.6%, 409 responses were eligible for analysis. The mean age of the participants (49.4% males and 50.6% of females) was 19.9±1.68 years. Baseline EHL scores were low, and there was no correlation with a demographic profile (all p<0.05, CI 95%). The demographic characteristics were similar between the two groups (for all, P <0.05, CI 95%) at baseline. The EHL scores improved in both groups between the pre-and post-test assessment; however, improvements were statistically significant only in the control group. The one-month post-intervention EHL scores were also higher in the intervention group compared to the control (p≤0.001, CI 95% for all). The total cost incurred for the intervention used was 11.5 USD.

**Data Availability Statement:** All relevant data are within the paper and its Supporting Information files.

**Funding:** No financial support was received for this study.

**Competing interests:** The authors have declared that no competing interests exist.

## Conclusion

Text message link demonstrated effectiveness for improving the EHL scores; the low baseline EHL scores substantially improved with intervention. The text message link intervention is a cost-effective method and could be considered in advocating for eye health in developing countries, particularly during global emergencies.

## Introduction

Global eye health is an emerging public health challenge of the 21st century. With the global population growing and aging, more people are developing and living with visual impairment (VI) [1]. VI poses a tremendous socioeconomic burden and affects the quality of life, ultimately plunging individuals into the vicious circle of poverty [2, 3]. The eye health progress is not keeping pace with needs, and we continue to face enormous challenges in elimination avoidable VI [4]. The World Report on Vision (WRV 2019) estimated that at least 2.2 billion people have VI, of which almost half could be prevented or has yet to be addressed. Uncorrected refractive error (43%) followed by unoperated cataract (33%), glaucoma (2%), and diabetic retinopathy (1%) are the leading causes of VI and are considered the priority ocular morbidities [5]. Literature suggests that with raised eye health awareness and provision of primary eye care services, 4/5th of VI are avoidable [5].

To improve universal eye health, the WRV recommends raising eye health awareness, engaging communities and empowering people about eye care needs. Literature acknowledges that the awareness and knowledge of common ocular morbidities are poor among the general population, causing a major barrier to uptake of eye care services [6–9]. The Low middle-income countries (LMICs) face ophthalmic human resources and financial constraints; providing eye health services and information to the population is a confronting task. Screening camps and awareness programs must meet to provide advocacy [10]. Elevating eye health literacy (EHL) within the key audiences plays a paramount role in eliminating avoidable blindness. During this COVID-19 pandemic, the need to identify and utilize cost-effective public health intervention is felt more than ever. In the LMICs, traditional ways of health education are time and resource-consuming and are least workable during pandemics. It calls for exploring the effectiveness of mobile health interventions that could aid health policymakers in planning interventions.

eHealth Technologies have emerged as an inexpensive, fast, and dynamic method of disseminating health information [11]. Besides with vast penetration and widespread reach, text messages with reliable links make it suitable for public health practices. However, with merely a few studies conducted to evaluate the effectiveness of text messages in health advocacy, the tool remains underutilized by public health professionals and policymakers [12–16]. Furthermore, eye health promotion has not received adequate priority [17]. With the ubiquity of mobile phone usage in a diverse population, mobile health interventions appear to be a promising medium for improving health education for all ages [18–20].

Considering the proliferation of mobile phones, and easy internet access amongst a diverse population, the study aimed to assess the effect of mobile health intervention (text message link) to promote EHL of priority eye diseases among university students. The hypotheses were; (1) the intervention group will have a higher EHL score after mobile health intervention (text message link) than of baseline, and (2) the intervention group will have a higher post intervention EHL score compared to the control group.

## Methods

### Design

This two-armed, parallel group (1:1) pre-test post-test questionnaire-based study evaluated the effect of mobile health intervention (text message link). Prior to commencement, study design and protocol got approved by the Institutional Ethical Committee (IRC) of the Amity University Haryana (AUH), Haryana, India (Reference: AUH/EC/D/2016/31). The study adhered to the tenets of the Declaration of Helsinki for human participants and followed the principles of Good Clinical Practice (GCP).

### Participants

Eligible participants were; (1) university students and (2) aged above 18 years. The study enrolled participants owning a smartphone with an active phone number and anticipating participation throughout the study. To reduce study bias, enrollment excluded students pursuing courses in optometry and visual sciences. Participants were invited through the university website and recruited between January and March 2018. It was implemented in three stages: baseline assessment, allocation and intervention, and post-intervention evaluation within six months (January to July 2018). On completing the study, participants received a free comprehensive eye examination at the Amity Optometry Clinic.

### Procedure

Data collectors attended a one-day training session covering necessary skills in data collection techniques, confidentiality, and privacy assurance. Participants fulfilling eligibility criteria were enrolled, informed that they were recruited for an interventional study and asked to sign an informed consent. Each participant was assigned a unique code to mask personal identifiers. Following enrollment, the socio-demographic information and web-enabled personal mobile phone number were recorded. Prior to randomisation and allocation, a EHL baseline assessment (pre-test) was conducted using a self-administered questionnaire. The intervention was assigned and the post-intervention assessment was conducted for both groups on the 30th day after the baseline assessment.

### Development of study questionnaire

A structured questionnaire was developed in English by the investigators [**S1 File**]. It was designed to assess the EHL (total scores on awareness and knowledge) of cataract, diabetic retinopathy, glaucoma, and refractive error. The questionnaire's face and content validity were assessed by faculties and reviewed by an expert panel of the university. The reproducibility and validity of the questionnaire were verified through a pilot study comprising 10% (n = 40) of the study population. The experience and feedback received from the pilot study were used to resolve the discrepancies and refine the questionnaire.

**Awareness.** The first question, which evaluated the individual's awareness, comprised whether the respondent had ever heard the name of the disease. Close-ended responses (yes or no) were recorded. Scores for yes and no were recorded as 1 and 0, respectively.

**Knowledge.** Responses to open-ended questions on symptoms and treatment options for each ocular condition were assessed. Providing at least one simple and correct symptom and treatment option of the disease was considered having knowledge; correct as 1 and incorrect as 0. Overall knowledge score was the total of scores on knowledge of symptoms and treatment options. The correct and incorrect responses participants provided are documented in **S1 Table**.

## Intervention

The intervention was a mobile health intervention (text message link) delivered as a text message. Two short text messages (SMS) were tailored; (1) a text message thanking participants for taking part in the baseline study (control text message) and, (2) a text message thanking participants for taking part in the baseline study along with a hyperlink (http://www.who.int/blindness/causes/priority/en/ ) (intervention text message) [**Fig 1**]. The hyperlink opened the WHO website on priority eye diseases containing a brief description of the background, causes, common symptoms, and treatment options for the priority eye diseases; cataract, glaucoma, diabetic retinopathy, and refractive error. The WHO website was chosen to provide reliable and consistent information about eye diseases. Participants allocated to control and intervention groups received the control text message and intervention text message, respectively, on the fifth day of baseline assessment.

## Outcome measures

All outcomes were self-reported and collected through a survey. The primary outcome measures included: (1) awareness, (2) knowledge, (3) changes in EHL (awareness and knowledge) scores for cataract, glaucoma, diabetic retinopathy, and refractive error from baseline to one-month post-test. Having heard about the disease was considered as having awareness and demonstrating some understanding about the symptoms and treatment options was considered having knowledge. The responses received from the participants are shown in S1 Table. The secondary outcome denoted the assessment of the cost-effectiveness of the intervention during the study period (presented separately).

## Sampling and sample size

An *a priori* power calculation was conducted to determine the sample size required. Assuming an effect size of 0.04 (small) between the two groups, and an alpha error of 0.05 (two-tailed), it required 328 participants to give power (1 - β) of over 95%. Considering a follow-up rate of 80%, and attrition at 15%, 442 questionnaires were distributed during the pre-intervention test. The sampling frame used student enrollment numbers from the university registry and participants selected by random selection, generated using Microsoft Excel.

| Control group | Dear [Name] |
|---|---|
| | Thank you for participating in the study on eye health literacy (EHL) on commom eye diseases. |
| **Intervention group** | **Dear [Name]** |
| | Thank you for participating in the study on eye health literacy (EHL) on commom eye diseases. To know more about eye diseases click |
| | http://www.who.int/blindness/causes/priority/en/ |

**Fig 1. Tailored text message for intervention and control group.**

### Randomization and blinding

After the baseline assessment (pre-test), the randomization was performed by the statistician. A list of participant numbers (unique to this study) was prepared, and computer-generated randomization allocated the participants to intervention and control groups in a 1:1 allocation sequence. The allocation was concealed from the participants, study staff, and investigators until the intervention was assigned. The data were collected by trained optometrists and analyzed by a statistician; both blinded to the intervention throughout the study. The participants and investigators delivering intervention could not be blinded due to the nature of the study.

### Data management and statistical analysis

Data coding, quality control, and data entry were done using established procedures. The questionnaires were pre-coded to minimize data coding errors. Before data entry, forms were checked for errors and necessary corrections made. Data were double entered by two different investigators using Epi-data version 3.1. Tools and checks of Epi-data software were used to control data entry errors and data cleaning performed.

Demographic characteristics of participants at baseline were compared using the Fisher exact test, Pearson chi-square test, and ANOVA. Descriptive tests were used to analyze baseline EHL assessment. Testing of hypothesis, for between-group changes in EHL scores at one-month, was performed using McNemar test and Wilcoxon matched-pairs test. All data analysis was two-sided at a 5% significance level and performed using the Statistical Package for Social Sciences (SPSS 21).

## Results

### Recruitment

A total of 450 students were assessed for eligibility; 428 recruited, 424 randomised and 409 response analysed. Using the 1:1 allocation approach, 424 were randomized into intervention and control groups equally (n = 212). After the post-intervention assessment, only 409 responses (203 control group and 206 intervention group) were eligible for the statistical analysis (response rate, 95.6%) The participant flow from enrollment through analysis is depicted in **Fig 2**.

### Baseline characteristics

The mean age of the participants (49.4% males and 50.6% females) was 19.9±1.68 (range 18–26) years. At the baseline, the demographic characteristics were similar between the two groups (for all, P <0.05, CI 95%). Notably, there were more undergraduate students and of Hindu ethnicity. There was no significant correlation between the EHL scores and demographic profile (all p<0.05, CI 95%). The baseline demographic characteristics of the study participants are summarised in Table 1. The baseline EHL scores of the study participants are summarized in **Fig 3**.

### Primary outcome

A statistical comparison found that both groups demonstrated better EHL (awareness and knowledge) scores at post-test compared to baseline assessment. **Table 2** shows the comparison of EHL score in the two groups, pre and post text message link intervention. The intervention group had a higher EHL score (p≤0.001, CI 95% for all) after mobile health intervention (text message link) than of baseline. This finding supported hypothesis 1.

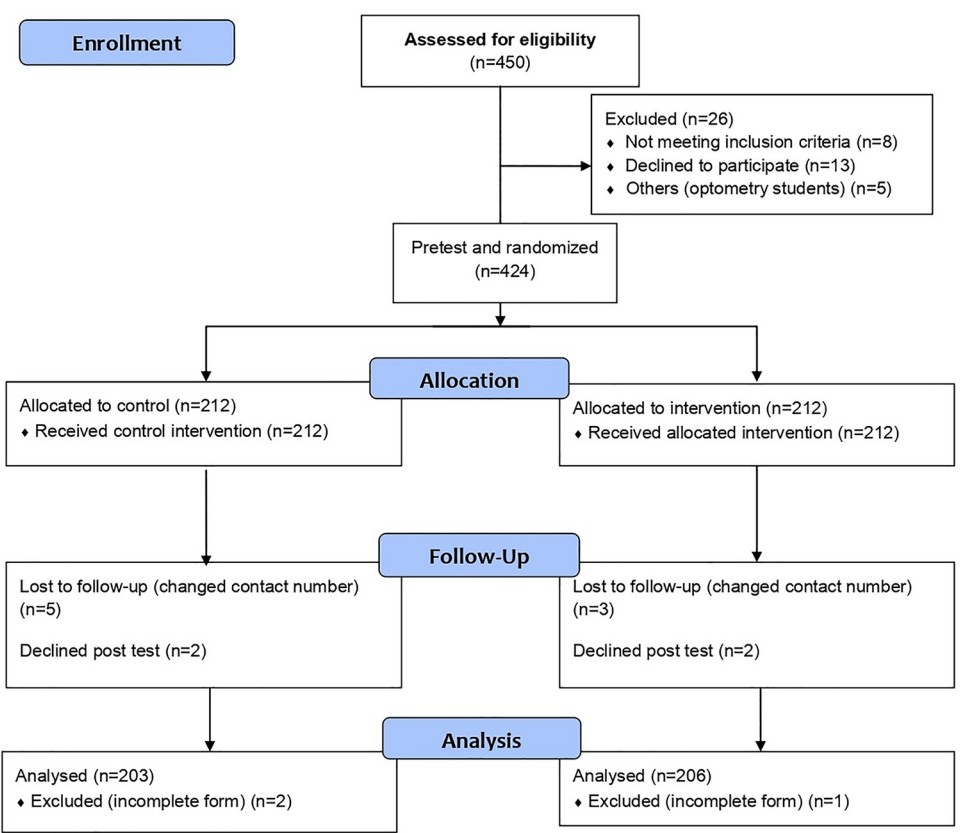

**Fig 2. The participant flow from enrollment through analysis.**

The changes in EHL scores between the pre-and post-test assessment between the two groups is shown in **Table 3**. At one-month post-test, the intervention group had a higher EHL score after intervention than that of the control group (p≤0.001, CI 95% for all). Thus, hypothesis 2 was supported.

## Secondary outcome

In the intervention group, 87.8% (181 of 206) participants responded they opened the hyperlink in the text message, while 83.4% (172 of 206) participants found it useful. Each text message costs us an average of INR 0.50; the total cost was INR 818 (USD 11.5).

## Discussion

Finding cost-effective methods for eye health promotion is an eye health priority, particularly in the LMICs. There is firm evidence that mobile phone messages can successfully promote healthcare, improve medication adherence, and change health behaviour [21–23]. This study has added fresh evidence supporting mobile health interventions (mobile text link) could effectively promote eye-health; the first of its kind to the best of investigators knowledge. Evidence of any effective eye health promotion methods that could benefit public health planning and advocacy is a crucial part of VISION 2020: Right to Sight [24]. With ubiquitous access and an increasingly popular communication platform even in the LMICs, text message tool is a well-established intervention for public health [25]. Recognizing that the number of characters in text messaging limits adequate dissemination of health information, this study aimed to

**Table 1. Homogeneity of demographic characteristics of the participants at the baseline.**

| Variables | | Control (1) n = 203 | Intervention (2) n = 206 | Total (n = 409) | P value |
|---|---|---|---|---|---|
| **Age (years± SD)** | | 19.77±1.58 | 20.01±1.77 | 19.9±1.68 | 0.155* |
| **Gender, n (%)** | | | | | |
| | Male | 99 (49.0) | 103 (51.0) | 202 | 0.843† |
| | Female | 104 (50.2) | 103 (49.8) | 207 | |
| **Religion, n (%)** | | | | | |
| | Hindu | 183 (50.3) | 181 (49.7) | 364 | 0.701*** |
| | Muslim | 7 (50.0) | 7 (50.0) | 14 | |
| | Christian | 8 (50.0) | 8 (50.0) | 16 | |
| | Sikh | 4 (40.0) | 6 (60.0) | 10 | |
| | Others | 1 (25.0) | 4 (75.0) | 5 | |
| **Course, n (%)** | | | | | |
| | Undergraduate | 182 (50.1) | 181(49.9) | 363 | 0.640† |
| | Postgraduate | 21 (45.7) | 25 (54.3) | 46 | |
| **Study Program, n (%)** | | | | | |
| | Arts | 36 (53.7) | 31(46.3) | 67 | 0.056‡ |
| | Business studies | 53 (57.6) | 39 (42.4) | 92 | |
| | Engineering | 45 (39.5) | 69 (60.5) | 114 | |
| | Science | 69 (50.7) | 67 (49.7) | 136 | |
| **Year of study, n (%)** | | | | | |
| | First | 71 (56.8) | 54 (43.2) | 125 | 0.059‡ |
| | Second | 74 (46.3) | 86 (53.7) | 160 | |
| | Third | 49 (51.6) | 46 (48.4) | 95 | |
| | Fourth | 9 (31.0) | 20 (69.0) | 29 | |
| **History of eye disease? n (%)** | | | | | |
| | No | 182 (49.3) | 187 (50.7) | 369 | 0.738‡ |
| | Refractive error | 18 (54.5) | 15 (45.5) | 33 | |
| | Diabetic retinopathy | 2 (66.7) | 1 (33.3) | 3 | |
| | Glaucoma | 1 (33.3) | 2 (66.7) | 3 | |
| | Cataract | 0 (0.0) | 1 (100) | 1 | |
| **History of eye disease in the family? n (%)** | | | | | |
| | No | 142 (52.2) | 130(47.8) | 272 | 0.253‡ |
| | Cataract | 13 (38.2) | 21 (61.8) | 34 | |
| | Glaucoma | 2 (100) | 0 (0.0) | 2 | |
| | Diabetic Retinopathy | 5 (62.5) | 3 (37.5) | 8 | |
| | Refractive error | 11 (50.0) | 11 (50.0) | 22 | |
| | Don't Know | 30 (42.2) | 41 (57.8) | 71 | |

SD, standard deviation.

*One way ANOVA between mean of groups.

† Fisher's exact test between group differences.

‡ Pearson Chi-square test between group differences.

capture if text message links could help mhealth interventions reach their full potential as a health advocacy strategy.

The baseline assessment suggests that the awareness of priority eye conditions was relatively poor amongst the study population; cataracts 68.7%, glaucoma 27.4%, diabetic retinopathy 24.9%, and refractive error 37.2%. Nevertheless, it is worthy to note that these findings align

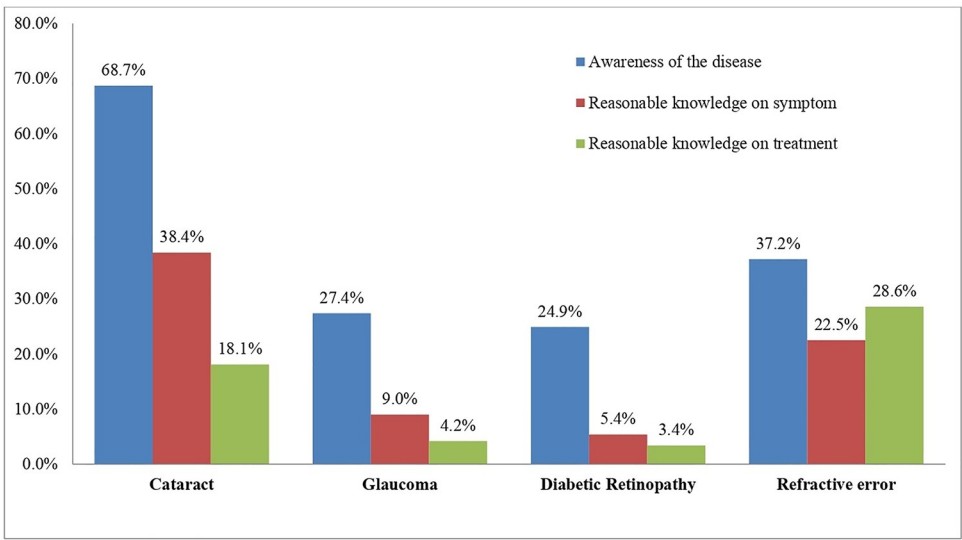

**Fig 3. The baseline EHL scores of the study participants.**

with the previous studies conducted in Asia [26, 27]. The knowledge of these common eye conditions is also poor, and correlates well with literature. A Low EHL score could mean inadequate or ineffective eye health advocacy programs in the community.

With the advancements in technologies over the past decade, a revolution has been occurring in health promotion [28]. In this study, the most notable finding was that, in the intervention group, the ESL scores after the intervention showed significant improvements for all four ocular conditions (p>0.05, CI 95%, for all). As hypothesized, EHL scores were significantly higher for intervention than that of the control group at one month. Though no studies were available to compare our findings on eye health, the results were consonant with the observations on other health care interventions [14, 15]. Text message link was also employed effectively as an online survey tool for longitudinal data collection [29]. The study hypotheses were

**Table 2. Changes in EHL (awareness and knowledge) scores between the pre-and post-test assessment between the two groups.**

| Disease | Outcome measure | Control Group n(%) | | | Intervention Group n(%) | | |
|---|---|---|---|---|---|---|---|
| | | Pre-test | Post-test | *p-value* | Pre-test | Post-test | *p-value* |
| Cataract | Awareness | 142 (67.0) | 142 (69.5) | 0.89 | 139 (65.6) | 153 (74.27) | <0.000 |
| | RKS | 86 (40.6) | 88 (42.0) | 0.50 | 71 (33.5) | 82 (39.8) | 0.001 |
| | RKT | 40 (18.9) | 48 (23.6) | 0.01 | 34 (16.0) | 48 (23.3) | <0.000 |
| Glaucoma | Awareness | 48 (22.6) | 50 (24.6) | 0.50 | 64 (30.2) | 95 (46.1) | <0.000 |
| | RKS | 19 (9.0) | 19 (9.4) | 0.97 | 18 (8.5) | 42 (20.3) | <0.000 |
| | RKT | 11 (5.2) | 15 (7.4) | 0.125 | 6 (2.8) | 24 (11.6) | <0.000 |
| Diabetic retinopathy | Awareness | 49 (23.1) | 55 (27.1) | .210 | 53 (25) | 85 (41.2) | <0.000 |
| | RKS | 12 (5.6) | 17 (8.4) | 0.063 | 10 (4.7) | 31 (15.4) | <0.000 |
| | RKT | 8 (3.8) | 12 (5.9) | 0.125 | 6 (2.8) | 28 (13.6) | <0.000 |
| Refractive error | Awareness | 77 (36.3) | 90 (44.3) | <0.000 | 75 (35.4) | 115 (55.8) | <0.000 |
| | RKS | 49 (23.1) | 56 (27.6) | 0.016 | 43 (20.3) | 67 (33.0) | <0.000 |
| | RKT | 54 (25.5) | 65 (32.0) | 0.080 | 63 (29.7) | 98 (47.5) | 0.001 |

RKS, reasonable knowledge of symptoms; RKT, reasonable knowledge on treatment.

**Table 3. The changes in EHL scores between the pre-and post-test assessment between the two groups.**

| Ocular Condition | Control Group n(%) | | | | Intervention Group n(%) | | | | (Time x Group) |
|---|---|---|---|---|---|---|---|---|---|
| | Baseline EHL score | Post-test EHL score | Mean difference (95% CI) | p-value | Baseline EHL score | Post-test EHL score | Mean difference (95% CI) | p-value | p-value |
| Cataract | 1.34±1.05 | 1.35±1.05 | 0.010 (0.00 to 0.02) | 0.16 | 1.18±1.03 | 1.23±1.08 | 0.04 (0.02 to 0.07) | 0.003 | 0.001 |
| Glaucoma | 0.38±0.78 | 0.39±0.76 | 0.010 (0.00 to 0.02) | 0.16 | 0.46±0.71 | 0.58±0.71 | 0.12 (0.07 to 0.16) | 0.000 | 0.001 |
| Diabetic Retinopathy | 0.36±0.76 | 0.37±0.77 | 0.010 (0.00 to 0.02) | 0.16 | 0.32±0.61 | 0.46±0.64 | 0.14 (0.09 to 0.18) | 0.000 | 0.001 |
| Refractive Error | 0.63±0.95 | 0.65±0.95 | 0.15 (0.00 to0.32) | 0.08 | 0.63±0.95 | 0.7±0.92 | 0.10 (0.06 to 0.14) | 0.000 | 0.003 |

EHL, eye health literacy; CI, confidence interval.

supported by the results, establishing that text message link intervention could effectively promote EHL.

The improvement in EHL in the group receiving text message link intervention could be attributed to several reasons. The mhealth technologies are portable and online materials can be easily accessed at individual's convenience with an internet connection. It allows the intervention to claim an individual's attention at the most convenient time by allowing temporal synchronization of the intervention delivery [30]. In 2019, more than half of the global population (4.13 billion) was connected to the internet via mobile phones [31]. The text message link intervention offers an additional advantage of not requiring to send multiple messages to disseminate a considerable amount of information. As more people use mobile phones throughout the day for various tasks, it is less likely to miss a text message prompt [32].

Although mobile health interventions are presently not utilized in public health for eye care, it was proven cost-effective as reminder programs [30]. Literature suggests a tremendous potential for text message links to positively affect public health intervention, particularly in the LMICs [33]. The cost-effectiveness of the method employed in this study can be established by the cost incurred (less than USD 12) to disseminate advocacy material through a hyperlink to the study population. This study provides shreds of evidence that sending text messages linked to a reliable website could be an effective medium for promoting EHL. While this study used English language-based text messages and a link to a English language website, interventions in vernacular languages could be even more cost-effective.

The COVID 19 pandemic has attacked the health system; ensuring access to health services is the cornerstone of successful health response [34]. During the COVID-19 pandemic more than ever, identifying and implementing effective health advocacy strategies is critical to enabling a better global response. A text message link is a tool to disseminate and reinforce information for eye health promotion effectively. Given its effectiveness, the text message link intervention could be considered a tool for equalizing access to information to address health disparities in minority populations.

## Limitation

While evidence in this study is largely in favour of text message link interventions, this study was not without limitations. The study cohort comprised the same university students, so the possibility of diffusion could not be eliminated. The generalization of the findings is also limited to university students. In the future, multicenter randomised controlled trials (RCTs) could address issues on how the program can be made more cost-effective. Registration of the study as a randomized controlled trial (RCT) was not possible within the stipulated time.

## Conclusion

The evaluation of mobile health intervention in this study provided evidence that text message links could improve EHL. The substantially low baseline EHL scores amongst educated individuals are of concern and call for the adoption of effective strategies for eye health promotion. Significantly higher EHL scores in the intervention group establish that text message links help promote eye health. Text message link intervention is also cost effective, and it could be used for promoting eye health, particularly by the LMICs during the current coronavirus pandemic.

## Supporting information

**S1 File. Questionnaire used for the study.**
(DOCX)

**S1 Table. Participant's responses on knowledge of priority eye disease during the baseline assessment.**
(DOCX)

**S1 Dataset.**
(XLS)

## Acknowledgments

We acknowledge Dr. Surinder Kumar Yadav, Department of Public Health, Amity Medical School, Amity University Haryana, India, for his guidance and support. We also thank the data collection team and participants for their time and cooperation.

## Author Contributions

**Conceptualization:** Indra Prasad Sharma, Monica Chaudhry, Raju Kaiti.

**Data curation:** Indra Prasad Sharma, Dhanapati Sharma.

**Formal analysis:** Indra Prasad Sharma, Monica Chaudhry, Dhanapati Sharma, Raju Kaiti.

**Investigation:** Indra Prasad Sharma, Dhanapati Sharma, Raju Kaiti.

**Methodology:** Indra Prasad Sharma, Monica Chaudhry, Dhanapati Sharma, Raju Kaiti.

**Project administration:** Indra Prasad Sharma, Monica Chaudhry.

**Resources:** Dhanapati Sharma.

**Software:** Indra Prasad Sharma, Dhanapati Sharma.

**Supervision:** Monica Chaudhry.

**Validation:** Indra Prasad Sharma, Monica Chaudhry, Raju Kaiti.

**Visualization:** Indra Prasad Sharma, Dhanapati Sharma.

**Writing – original draft:** Indra Prasad Sharma, Dhanapati Sharma, Raju Kaiti.

**Writing – review & editing:** Indra Prasad Sharma, Monica Chaudhry, Dhanapati Sharma, Raju Kaiti.

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
