## [Decision Letter · Decision Letter 0]

12 Jul 2021

 PGPH-D-21-00077 Mobile health intervention for promotion of eye health literacy PLOS Global Public Health   *Decision: Major Revision* 

Dear Dr. Sharma,

Thank you for submitting your manuscript to PLOS Global Public Health. After careful consideration, we feel that it has merit but does not fully meet PLOS Global Public Health’s publication criteria as it currently stands.   Therefore, we invite you to submit a revised version of the manuscript that addresses the points raised during the review process.

We look forward to receiving your revised manuscript.

Kind regards,

Nick Drydakis, Ph.D

Academic Editor

Reviewer's comments:

Reviewer's Responses to Questions

**Comments to the Author**

1. Does this manuscript meet PLOS Climate’s publication criteria? Is the manuscript technically sound, and do the data support the conclusions? The manuscript must describe methodologically and ethically rigorous research with conclusions that are appropriately drawn based on the data presented.

Reviewer #1: Yes

2. Has the statistical analysis been performed appropriately and rigorously?

Reviewer #1: Yes

3. Have the authors made all data underlying the findings in their manuscript fully available (please refer to the Data Availability Statement at the start of the manuscript PDF file)?

Reviewer #1: Yes

4. Is the manuscript presented in an intelligible fashion and written in standard English?

PLOS Climate does not copyedit accepted manuscripts, so the language in submitted articles must be clear, correct, and unambiguous. Any typographical or grammatical errors should be corrected at revision, so please note any specific errors here.

Reviewer #1: Yes

5. Review Comments to the Author

Reviewer #1: The authors present a well written article documenting the use of text messaging as a means of improving eye health literacy in a low middle-income country. They clearly demonstrate an increased awareness of eye related issues in the participants in the intervention group.

However, there are certain questions that need to be answered.

Did the participants know they were in an interventional study (although they might not know what the intervention was)?

Did they know they would receive a Free Eye Exam at the time or enrollment?

Add in the key for the demographic data needs to be added just for clarity.

As the authors pointed out they selected a very defined population of young, presumably English speaking, college students. Such a population does not necessarily reflect the target population.

Another issue is the language used. The authors used an English language-based text message and a link to a English language website. Would this be an issue when dealing with a diverse population such as in India, especially from a public health perspective? Many of the target populations may not be fluent/comfortable with English.

Any thoughts on the use of vernacular languages??

Did any of the participants or their family have any eye issues other than those listed in the paper?

Are smartphones as a means of public health measures, feasible in the rural parts of a country?

I believe the authors touched up on this but is it possible that some students talked to each other perhaps two people, one of whom got the link and one who did not?

Can the increased EHL scores be due to the general increased interest since they were participating in clinical trial or the intervention itself?

Were there any differences in sub-group analysis, e.g. within the intervention group did the Science majors do better than Arts Majors?

6. PLOS authors have the option to publish the peer review history of their article (what does this mean?). If published, this will include your full peer review and any attached files.

**Do you want your identity to be public for this peer review?** For information about this choice, including consent withdrawal, please see our Privacy Policy.

Reviewer #1: No

---

## [Editor Report · Decision Letter 1]

15 Sep 2021

Mobile health intervention for promotion of eye health literacy

PGPH-D-21-00077R1

Dear Dr. Sharma,

We're pleased to inform you that your manuscript has been judged scientifically suitable for publication and will be formally accepted for publication once it meets all outstanding technical requirements.

Within one week, you'll receive an e-mail detailing the required amendments. When these have been addressed, you'll receive a formal acceptance letter and your manuscript will be scheduled for publication.

An invoice for payment will follow shortly after the formal acceptance. To ensure an efficient process, please log into Editorial Manager at https://www.editorialmanager.com/pgph/ click the 'Update My Information' link at the top of the page, and double check that your user information is up-to-date. If you have any billing related questions, please contact our Author Billing department directly at authorbilling@plos.org.

Kind regards,

Nick Drydakis, Ph.D

Academic Editor

Additional Editor Comments (optional):

In the current version of the manuscript, the authors have addressed the reviewer’s comments and suggestions.